# Structural Insights on Hyp-Gly-Containing Peptides as Antiplatelet Compounds through Topomer CoMFA and CoMSIA Analysis

**DOI:** 10.3390/foods12040777

**Published:** 2023-02-10

**Authors:** Yijie Yang, Qi Tian, Shiming Li, Bo Li

**Affiliations:** 1Beijing Laboratory for Food Quality and Safety, College of Food Science and Nutritional Engineering, China Agricultural University, Beijing 100083, China; 2Key Laboratory of Precision Nutrition and Food Quality, Department of Nutrition and Health, China Agricultural University, Beijing 100083, China

**Keywords:** antiplatelet, collagen peptide, thrombosis, QSAR, Topomer CoMFA, CoMSIA

## Abstract

Increasing evidence has shown collagen hydrolysate involves a variety of bioactivities. In our previous study, multiple antiplatelet peptides containing Hyp/Pro-Gly were identified in collagen hydrolysates from Salmo salar and silver carp skin and exhibited anti-thrombosis activity without bleeding risks in vivo. However, the relationship between structure and activity remains unknown. We performed 3D-QSAR studies on 23 Hyp/Pro-Gly-containing peptides in which 13 peptides were reported before. CoMFA, Topomer CoMFA and CoMSIA analyses were used to generate the QSAR models. Topomer CoMFA analysis showed a q^2^ value of 0.710, an r^2^ value of 0.826, an r^2^_pred_ value of 0.930, and the results showed that Hyp instead of Pro was more important for improving the antiplatelet activity. CoMSIA analysis showed a q^2^ value of 0.461, an r^2^ value of 0.999, and an r^2^_pred_ value of 0.999. Compared with the electrostatic field and hydrogen bond donor field, the steric field, hydrophobic field and hydrogen bond receptor field have great influence on the activity of antiplatelet peptides. The predicted peptide EOGE exhibited antiplatelet activity induced by ADP, and inhibited thrombus formation (300 μmol/kg bw) without bleeding risks. Combined results of these studies indicate that OG-containing peptides had a potential to be developed into an effective specific medical food in the prevention of thrombotic diseases.

## 1. Introduction

Thrombosis is the leading cause of mortality and morbidity worldwide and participates in the pathogenesis of numerous cardiovascular diseases (CVDs), such as stroke, acute myocardial infarction and sudden cardiac death [1,2]. The platelet has long been assumed as a therapeutic target in thrombosis diseases for its role in hemostasis and in the initiation and propagation of thrombus formation [3]. Accordingly, the development of antiplatelet drugs that block platelet activation and aggregation will provide excellent therapeutic strategies to treat and prevent thrombotic diseases clinically. However, current antiplatelet drugs are still limited for their side effects, especially bleeding complications [4]. Thus, it is essential to develop natural antiplatelet compounds for diet intervention of CVDs since this offers fewer side effects and lower cost.

Collagen is a natural source of bioactive peptides and widely distributed in skin and bone tissue, which has a unique triple-helix structure with a repeated amino acid sequence of (Gly-X-Y)n, in which X and Y are typically Pro and Hyp [5]. Hyp-Gly-containing peptides are abundant in collagen, and a large quantity of these peptides could be released after hydrolysis. In our previous study, multiple collagen peptides containing a Hyp-Gly motif separated and identified from *Salmo salar* and *Hypophthalmichthys Molitrix* skin were found to exhibit antiplatelet aggregation activity induced by ADP, and resist gastrointestinal digestion to be absorbed intact [6,7,8,9]. Additionally, oral administration of the peptide OGEFG (OG-5) identified from *Salmo Salar* skin also attenuates thrombus formation without bleeding risks in vivo [10]. Fortunately, most of the Hyp-Gly-containing peptides were also successfully enriched by macroporous resin from collagen hydrolysates [11]. These results reveal that the identified OG-containing peptides in collagen hydrolysates may have potential to be developed as an effective diet supplement to prevent the occurrence of thrombotic disease. However, the relationship between the structure and activity of antiplatelet peptides remains unknown.

Quantitative structure–activity relationship (QSAR) is a common research method to explore the relationship of molecular structure and its biological activity. The 3D-QSAR model has become a popular research method that can be used to predict the structure of compounds with high activity [12]. Recently, computer-aided drug design based on the 3D-QSAR model has been widely used in the research and development of a variety of peptides, such as antibacterial peptides, antihypertensive peptides and bitter-tasting peptides [13,14,15]. In CoMFA, steric and electrostatic field descriptors were calculated with the distant-dependent dielectric constant. CoMSIA was similar to CoMFA about the fields around the aligned molecules with three other descriptors named hydrophobic, hydrogen-bond donor and hydrogen-bond acceptor calculated together [16]. In 2003, Cramer R.D. first combined CoMFA and Topomer to develop the 3D-QSAR model—Topomer CoMFA, which is more accurate and efficient than CoMFA and CoMSIA [17]. Topomer CoMFA can predict the specific activity of a series of compounds with a common skeleton through the structure of R-groups [18]. In recent years, many studies have used the Topomer CoMFA model to investigate the rules between the structure of drug molecules and natural product molecules and their activities [19,20,21,22].

In this study, Topomer CoMFA and CoMSIA analyses were employed to establish the 3D-QSAR model to predict the structure features of antiplatelet peptides based on the structure and pIC_50_ (pIC_50_ = –logIC_50_) values of these antiplatelet peptides. The predicted peptide was synthesized to investigate in vitro antiplatelet aggregation activity and in vivo anti-thrombosis activity. This study will make contribution to the ongoing research on antiplatelet peptides in natural compounds and its application in selective hydrolysis.

## 2. Materials and Methods

### 2.1. Material and Chemicals

Adenosine diphosphate (ADP) was purchased from Solarbio (Beijing, China). Clopidogrel hydrogen sulfate were purchased from MCE (Shanghai, China). Phosphate buffer solution (PBS) was purchased from HyClone (Beijing, China). Peptide was synthesized with purity over 98% by GL Biochem Ltd. (Shanghai, China). Other chemicals were all analytically pure grade and purchased from Sinopharm chemical reagent Co., Ltd. (Beijing, China).

### 2.2. Feeding Conditions and Animals

Eight-week-old male SD rats (180 ± 5 g, SPF grade) were purchased from Beijing Vital River Experiment Animal Technology Co., Ltd. (Beijing, China). The rats were acclimatized for a week in pathogen-free animal housing under a 12-h day and night cycle at 21 °C with access to a regular chow diet and tap water. All experiments involving animals were performed in compliance with the relevant laws and institutional guidelines with the Welfare Committee of the Centre of Experimental Animal, Beijing, China (Approval No. AW01080202-1).

### 2.3. Antiplatelet Activity Assay In Vitro

The platelet was isolated from rat aorta blood and mixed with sodium citrate (1:9, v/v) as previously described with some modifications [23]. Briefly, anticoagulated blood was centrifuged (50× *g*, 10 min) to obtain platelet-rich plasma (PRP). The residue was centrifuged at 750× *g* for 10 min to collect the platelet-poor plasma (PPP), and PRP was diluted to 2–3 × 10^8^ with PPP.

Inhibition of platelet aggregation induced by ADP was measured with a four-channel aggregometer as previously described [24,25]. Aliquots of 270 μL of PRP were incubated with 30 μL of PPP or peptides with different concentrations at 37 °C for 5 min. After incubation, platelet aggregation was induced by adding 30 μL of ADP (0.1 mM). The platelet aggregation at 5 min was recorded, and the inhibition rate was calculated as follows.
Inhibition of platelet aggregation (%) = [1 − platelet aggregation (sample)/platelet aggregation (control)] × 100

The IC_50_ value of platelet aggregation was calculated by Graphpad Prism 6.0.

### 2.4. Data Sets

Thirteen antiplatelet collagen peptides containing OG(PG) sequence identified from collagen hydrolysates, reported in our previous study, plus ten peptide sequences were selected from collagen amino acid sequences (NCBI database) and synthesized. A total of twenty-three antiplatelet aggregation peptides containing OG(PG) sequence identified from collagen hydrolysates, reported in our previous study, were used for generation of all the QSAR models using Sybyl-X 2.0 software (Tripos Corporation, USA). Table 1 includes chemical structures of all peptides, along with their IC_50_ and pIC_50_ values, which were used to perform 3D-QSAR analysis.

### 2.5. Topomer CoMFA Analysis

Topomer CoMFA is an alignment-independent 3D-QSAR method that correlates topological, electrostatic and steric properties of molecules with biological activity [17,26]. The Topomer CoMFA model was developed by splitting the compounds into fragments, topomerically aligning each fragment and calculating steric and electrostatic fields at a regular space grid of 2 Å. The steric and electrostatic descriptors of these fragments were obtained and calculated using a default energy cut-off of 30 kcal/mol. A probe CH_4_ with +1 charge and Van der Waals radius of 1.0 Å was employed. PLS regression was used for generating the Topomer CoMFA with leave-one-out (LOO) cross-validation analysis.

### 2.6. CoMSIA Analysis

The 3D-QSAR study of the compound was also carried out by using the CoMSIA method in Sybyl-X 2.0 (Tripos, USA), and the scores of the five fields including the three-dimensional field, the electrostatic field, the hydrophobic field, the hydrogen bond donor field, and the hydrogen bond acceptor field were calculated. The relevant parameters were set as follows: the stereo field adopts the CH_4_ probe, hydrophobicity +1.0, charge +1.0 valence, van der Waals radius 1.0 Å, hydrogen bond donor strength +1, hydrogen bond acceptor strength +1, and other parameters were system default values. The Gaussian function was used to calculate the distance of the CH_4_ probe molecule to each atom in the antiplatelet peptide without specifying an energy cut-off.

### 2.7. Ferric Chloride-Induced Arterial Thrombosis Model

To estimate the effects of the peptides on thrombus formation in vivo, a ferric chloride-induced arterial thrombosis model in rats was used according to the described method with some modifications [27]. SD rats were randomly divided into 4 groups (*n* = 6) and orally administrated with the vehicle (saline), peptide EOGE (200 and 300 μmol/kg, bw) and clopidogrel (65 μmol/kg, bw), respectively. After 1 h, the vascular injuries were induced by applying a filter paper (5 × 10 mm) saturated with 25% ferric chloride on the adventitial surface of the exposed left common carotid arteries. The injured carotid artery segment was excised, and the thrombus was washed carefully with PBS. After blotting the excess liquid, the thrombus was weighed immediately.

### 2.8. Determination of Thymus Index and Spleen Index

After the rats were sacrificed, the thymuses and spleens were collected and weighed to obtain the thymus index (TI) and spleen index (SI). The TI and SI were calculated according to the following equation: TI/SI (mg/g) = (weight of thymus/spleen)/body weight.

### 2.9. Coagulation Parameters

The prothrombin time (PT), activated partial thromboplastin time (APTT), and thrombin time (TT) assay methods refer to our previous study [28]. PT, APTT, and TT detection kits (Siemens, Shanghai, China) were employed for the determination of coagulation parameters according to the manufacturer’s instructions.

### 2.10. Statistics Analysis

Results were expressed as the mean ± SD of at least three independent experiments. The statistical significance was assessed by one-way analysis of variance (ANOVA) followed by Duncan’s test using SPSS 19.0. Differences were considered statistically significant when *p* < 0.05.

## 3. Results

### 3.1. Establishment of 3D-QSAR Model

In our previous study, a series of OG-containing peptides were identified from collagen hydrolysates in fish skin and exhibited potent inhibitory activity against platelet aggregation [6,7]. Topomer CoMFA, CoMFA and CoMSIA analyses were employed to establish the 3D-QSAR model of antiplatelet peptides. All statistical parameters of QSAR models were shown in Table 2. The higher q^2^, r^2^ and r^2^_pred_, and F-values, along with the lower SEE value, indicate a better statistical correlation and reasonable predictability of the QSAR model. The optimal number of principal components of the model established by Topomer CoMFA is 2. The regression correlation coefficient r^2^ = 0.862 and the LOO cross-validation correlation coefficient q^2^ = 0.710, and the external validation regression correlation coefficient r^2^_pred_ = 0.930. These results indicate that the Topomer CoMFA model has good predictive ability. Similarly, for the CoMSIA model, the regression correlation coefficient r^2^ = 0.999 and the cross-validation correlation coefficient q^2^ = 0.461. The normalized correlation coefficient r^2^_pred_ = 0.999, indicating that the external prediction ability of the model is extremely high. However, the q^2^ and r^2^_pred_ values of the CoMFA model were <0.4 and 0.8, respectively, indicating its poor predictability. Meanwhile, the CoMFA model exhibits poor stability, whose SEE is 0.091 and the F value of Fisher’s test is 67.769. Therefore, the model established by CoMFA cannot reflect the relationship between the structure and activity of antiplatelet peptides. The Topomer CoMFA and CoMSIA models were employed to predict the antiplatelet aggregation activity of peptides.

### 3.2. Topomer CoMFA Analysis of OG(PG)-Containing Peptides

To establish the Topomer CoMFA model of antiplatelet peptides, Pro-Gly was designed as the common skeleton. The amino acid sequence of anti-platelet peptides was cleaved into a common skeleton PG and three R groups (R1, R2 and R3). The R1 group is the hydroxyl group on hydroxyproline or the hydrogen atom on the proline, the R2 group is the N-terminal amino acid residues, and the R3 group is the C-terminal amino acid residues (Figure 1A). The 23 antiplatelet peptides (Table 1) in the training and test sets were aligned, after the processing of energy minimization (Figure 1B). Then the steric and electrostatic fields of the fragments were calculated and partial least-squares (PLS) regression was used for generating the Topomer CoMFA with leave-one-out (LOO) cross-validation analysis. The activity of antiplatelet peptides in the model established by Topomer CoMFA were close to the Y = X regression line (r^2^ = 0.862), indicating that there was little deviation between the measured pIC_50_ and the predicted pIC_50_ (Figure 2A,B).

Take, as an example, the tetrapeptide ROGE with R1, R2 and R3 groups as a typical structure, whose steric and electrostatic contour maps are shown in Figure 2C–H. Steric and electrostatic contour maps can intuitively reveal the calculation results of the model and explain the effect of changes in the local structure of the compound on the overall activity of the molecule [19]. In the steric contour maps, if the larger groups are introduced into the green region, or the smaller groups into the yellow region, the obtained derivative would have better antiplatelet activity. In the electrostatic contour maps, blue contours represent areas at which electropositive groups improve activity, and red contours show that electronegative groups improve activity [29].

Obviously, in the steric contour maps of R1 (Figure 2C), the H atom of the hydroxyl group was located in the green area. Therefore, for these PG-containing antiplatelet peptides, whether their proline is hydroxylated affects their activity positively. For the R2 group (the N-terminal of the common skeleton) (Figure 2E), there were two yellow regions near the connection site, and a green region at the far end, which means the R2 group with large steric hindrance (such as Phe, Tyr, Trp, Pro and Hyp) would reduce the activity. Furthermore, the introduction of amino acids with long molecular configurations (such as Glu, Asp) would increase the activity. Similarly, in the steric contour maps of the R3 group (the C-terminus of the common skeleton) (Figure 2G), there were two yellow areas near the access site and a large green area at its far end, indicating the introduction of amino acids with long molecular configurations (such as Glu, Lys, Asp, His, Gln and dipeptide Ser-Ala) would affect the activity positively. Compared to R2, the yellow region of R3 is closer to the common skeleton PG. Moreover, the green region of R3 is larger than that of R2. Therefore, the absence of the R3 group would significantly reduce the activity of whole molecule. These results were consistent with the activity contribution levels of different groups in the data table of the Topomer CoMFA analysis (Appendix A).

In the electrostatic contour maps (Figure 2D,F,H), the blue region is favorable for electropositive groups, and red regions are favorable for electronegative groups. For the R1 group (Figure 2D), there was a red area near the oxygen atom and a blue area near the access site, indicating that the introduction of the electronegative group hydroxyl would increase the activity. Therefore, the activity of OG-containing peptides was generally higher than that of PG-containing peptides, which is consistent with the results of the steric contour maps. For the R2 group (Figure 2F), there were some scattered red areas (40%) near the access site and a huge blue area (60%) at the far end, indicating that the introduction of electropositive amino acids (Lys, Arg, and His) was favorable for the activity. For the R3 group (Figure 2H), the red area and the blue area were both located near the access site and were similar in size, so the introduction of electropositive or electronegative amino acids had no significant effect on the activity. Thus, the effect of the steric field on the activity was more significant than that of the electrostatic field.

According to the results of the Topomer CoMFA analysis (Figure 2I,J), introduction of an acidic amino acid (Glu) or a basic amino acid (Arg) will improve the overall activity of the molecule. Combination of Glu and Arg to N-terminal residue exhibits a similar result. The connection of hydrophilic amino acids including Q, R, E, H, S, and K to the C-terminal also has a positive effect on antiplatelet aggregation activity.

### 3.3. CoMISA Analysis of OG(PG)-Containing Peptides

The CoMSIA model concentrates on the effects of the steric field, electrostatic field, hydrophobic field, hydrogen bond acceptor field, and hydrogen bond donor field on the activity of the compounds. In addition, the investigation of the molecular hydrogen-bond donor field and hydrogen-bond acceptor field by the CoMSIA method is helpful for analyzing the binding mode between compounds and receptor, which is helpful for the search for the binding site. The linear regression curve of the CoMSIA model between measured pIC_50_ and predicted pIC_50_ is shown in Figure 3A; almost all antiplatelet peptides in the model established by CoMSIA are located on the Y = X regression line, indicating an excellent correlation between the measured pIC_50_ and the predicted pIC_50_.

The 3D equipotential diagram of ROGE of the QSAR model using CoMSIA is shown in Figure 3B–F. The steric contour map and electrostatic contour map in the CoMSIA model are similar to the results in Topomer CoMFA. Interestingly, the electrostatic contour map shows that there is no red region but two blue regions at the C-terminal (Figure 3C), indicating that the introduction of two positively charged groups here will have a positive effect on the activity of the peptide. This result is consistent with the positive contribution of the R3 group of the tetrapeptide PGHH in the Topomer CoMFA model.

In the hydrophobic contour map, the orange region indicates that the increase in the hydrophobic group is beneficial to the antiplatelet aggregation activity of the peptide. As shown in Figure 3D, a large orange region is observed at the N-terminal of ROGE, indicating that a non-polar amino-acid-containing hydrophobic group at the N-terminal would strengthen the activity.

In the hydrogen bond donor contour map, the blue region indicates that the activity of the peptide enhanced with the increase of hydrogen bond donors, which is opposite to that of the purple region. As shown in Figure 3E, two regions are concentrated in the common skeleton, suggesting that the hydrogen bond donor map has little effect on the antiplatelet activity of peptides.

Similarly, pink regions in the hydrogen bond acceptor map indicates that the increase in hydrogen bond acceptors strengthens the antiplatelet activity, which is totally opposite to that of green regions. As shown in Figure 3F, the pink regions are small, while the green regions are widely distributed in the common skeleton and the far ends of the N-terminal and C-terminal. These results suggested that a long main chain with a quantity of hydrogen atoms, such as Arg, would contribute to a higher antiplatelet activity.

According to the results of the CoMSIA analysis, the contribution value of each field is shown in Figure 3G. The steric contour map, hydrophobic contour map and hydrogen bond acceptor contour map exhibit a greater effect on the activity of antiplatelet peptides.

### 3.4. Tetrapeptide EOGE Inhibited Platelet Aggregation In Vitro and Thrombosis In Vivo

According to the 3D-QSAR model, a tetrapeptide EOGE which has a relatively high predicted activity was designed and synthesized for assay of the antiplatelet activity. Peptide EOGE was also searched in collagen amino acid sequences, and it was found that it locates at the 192–195 site of α1 chain from skin type I collagen of *Hypophthalmichthys Molitrix.* As shown in Figure 4A, EOGE inhibited ADP-induced platelet aggregation in a dose-dependent manner. The IC_50_ value of EOGE was 0.39 mM, which was higher than the inhibition activity of the identified peptides (OGSA with 0.63 mM and OGE with 0.62 mM) and the model prediction value of 0.45 mM (data not shown). To evaluate the anti-thrombosis activities of EOGE in vivo, a ferric chloride-induced arterial thrombosis model in rats was employed. As shown in Figure 4B, oral administration of EOGE inhibited the thrombus formation significantly at a dose of 300 μmol/kg b.w. (*p* < 0.01). However, no significant increase of anti-thrombosis activity was observed in vivo when compared with that of the peptide OGE, which might be due to partial degradation in the gastrointestinal tract.

### 3.5. Effect of EOGE on Coagulation Parameters

The coagulation parameters after oral administration of EOGE were determined to evaluate the bleeding risk. As shown in Figure 5A, oral administration of EOGE at dosages of 200 μmol/kg b.w. and 300 μmol/kg b.w. had no significant effect on coagulation parameters, including PT, TT, and APTT. However, clopidogrel altered the TT and PT significantly. This result indicates that EOGE had no bleeding risk at the effective dosage, and its bleeding risk was much lower than that of clopidogrel.

### 3.6. Effect of EOGE on the Thymus Index and Spleen Index

The TI and SI after peptide treatment were measured to preliminarily investigate whether the peptide dose had obvious toxicological effects on the animal subjects [30]. As shown in Figure 5B, the TI and SI of the EOGE- and clopidogrel-treated groups had no significant difference compared to that of the vehicle group. There was no obvious atrophy and hyperplasia or swelling of thymus and spleen. This result demonstrates that oral administration of the EOGE at dosages of 200 or 300 μmol/kg b.w. had no acute toxicological effects.

## 4. Discussion

In this study, the 3D-QSAR model of Hyp/Pro-Gly-containing peptides was established by Topomer CoMFA and CoMSIA. Topomer CoMFA only investigated the effects of stereoscopic field and static electric field on the activity of the compounds, while the model established by CoMSIA investigated the effects of the stereoscopic field, electrostatic field, hydrophobic field, hydrogen bond receptor field, and hydrogen bond donor field on the activity of compounds. However, the principles of these two methods are different. Topomer CoMFA scores the stereoscopic and static electric fields of different R groups through the fragment of active molecules to obtain the contribution value, and output the three-dimensional equipotential diagram of the R group. The influence of each R group on the activity of the compounds can be analyzed in detail, so as to obtain the key structural characteristics of highly active molecules. CoMSIA evaluates the fields of each molecule, finally obtains the contribution value of the five fields and three-dimensional equipotential diagram. Its advantage is that the output of the three-dimensional equipotential diagram of the whole molecule is convenient for the analysis of the whole molecule, and the fields around the common skeleton will also be displayed. It can be considered to optimize and transform the common skeleton to add other advantages of R bases. In addition, the investigation of the molecular hydrogen bond donor field and hydrogen bond acceptor field by the CoMSIA method is conducive to the analysis of the molecular and receptor binding mode, suggesting that the ligand easily forms a hydrogen bond position, which is conducive to the search of binding sites.

For the two 3D-QSAR models established by Topomer CoMFA and CoMSIA obtained in this study, although their statistical parameters meet the requirements of excellent models and the key structural characteristics of highly active antiplatelet peptides are pointed out, the N-terminal and C-terminal conditions are not comprehensive enough due to the small number of model samples. A large number of certain amino acids, such as Glu and Arg, appeared in the database, while Asn, Cys, Asp, and other amino acids, which accounted for a relatively small proportion in the collagen sequence, were not investigated. On the other hand, the longest peptide in the database was heptaeptide, and no more than three amino acids were connected to the N-terminal and C-terminal of the common skeleton. Therefore, the study could not explain the effect of the number of amino acid residues connected at both ends on the activity. This also provides a direction for future research to expand the range of sequence sources of Hyp/Pro-containing peptides beyond fish skin collagen to provide a more comprehensive explanation of the key structural features of antiplatelet peptides with high activity.

This study also demonstrated that the antiplatelet peptide activity is strong in the range of tetrapeptide to heptapeptide. However, the content of the long-chain peptide is very low after oral administration, and it is almost impossible for the specific sequence to exist. The structure of the R group can be modified to become a peptide compound or peptide derivatives, so as to improve its stability in plasma to reach the target and improve its antiplatelet activity.

## 5. Conclusions

In this study, Topomer CoMFA and CoMSIA analysis were employed to establish a 3D-QSAR model to predict the structure features of antiplatelet peptides. The overall required structures were predicted from the contour map analysis from Topomer CoMFA and CoMSIA to design highly active antiplatelet peptides. Based on the Topomer CoMFA and CoMSIA analyses of 23 antiplatelet peptides, characteristics of highly active antiplatelet peptides can be obtained: the common skeleton is a Hyp-Gly sequence, and the N-terminus is connected to charged long-chain amino acid residues, and the C-terminus is connected to a hydrophilic, large sterically hindered amino acid residue. Suitable amino acids for the N-terminus are: R, E, H, K, and D, which are also suitable for the C-terminal. Additionally, G, S, T, C, Y, N, and Q are suitable for the C-terminal. These newly found characteristics will provide insight into the structure requirement for the development of antiplatelet peptides and highlight the potential application of the OG-containing peptides as dietary supplements to prevent cardiovascular diseases by inhibiting platelet aggregation.

## Figures and Tables

**Figure 1 foods-12-00777-f001:**
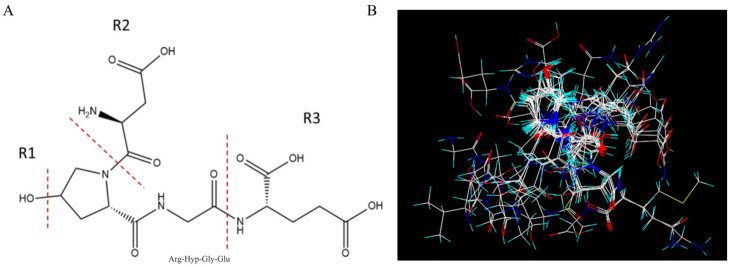
Establishment of 3D-QSAR model. (**A**) The common skeleton OG(PG) and R1, R2, R3 groups. (**B**) Molecular alignment of 23 antiplatelet peptides.

**Figure 2 foods-12-00777-f002:**
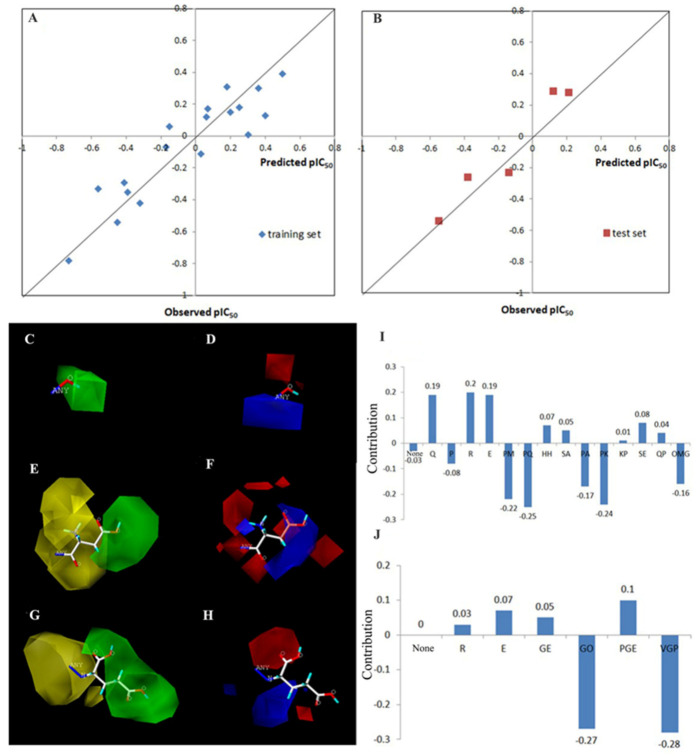
The 3D-QSAR model established by Topomer CoMFA. Plots of predicted values of pIC_50_ versus actual values for (**A**) training set (r^2^ = 0.862) and (**B**) test set (r^2^ = 0.930) of antiplatelet peptides using the Topomer CoMFA model. (C-H) Steric and Electrostatic Contribution maps for tetrapeptides ROGE by Topomer CoMFA analysis. The color of ROGE molecular diagram: White-C, Blue-N, Aquamarine-H, Red-O, Yellow-S. ANY: The sites where the R groups are introduced to the PG common skeleton. (**C**) Steric Contribution map of R1 group. (**D**) Electrostatic Contribution map of R1 group. (**E**) Steric Contribution map of R2 group. (**F**) Electrostatic Contribution map of R2 group. (**G**) Steric Contribution map of R3 group. (**H**) Electrostatic Contribution map of R3 group. (**I**) Total Contribution of C-terminal amino acid residues. (**J**) Total Contribution of N-terminal amino acid residues.

**Figure 3 foods-12-00777-f003:**
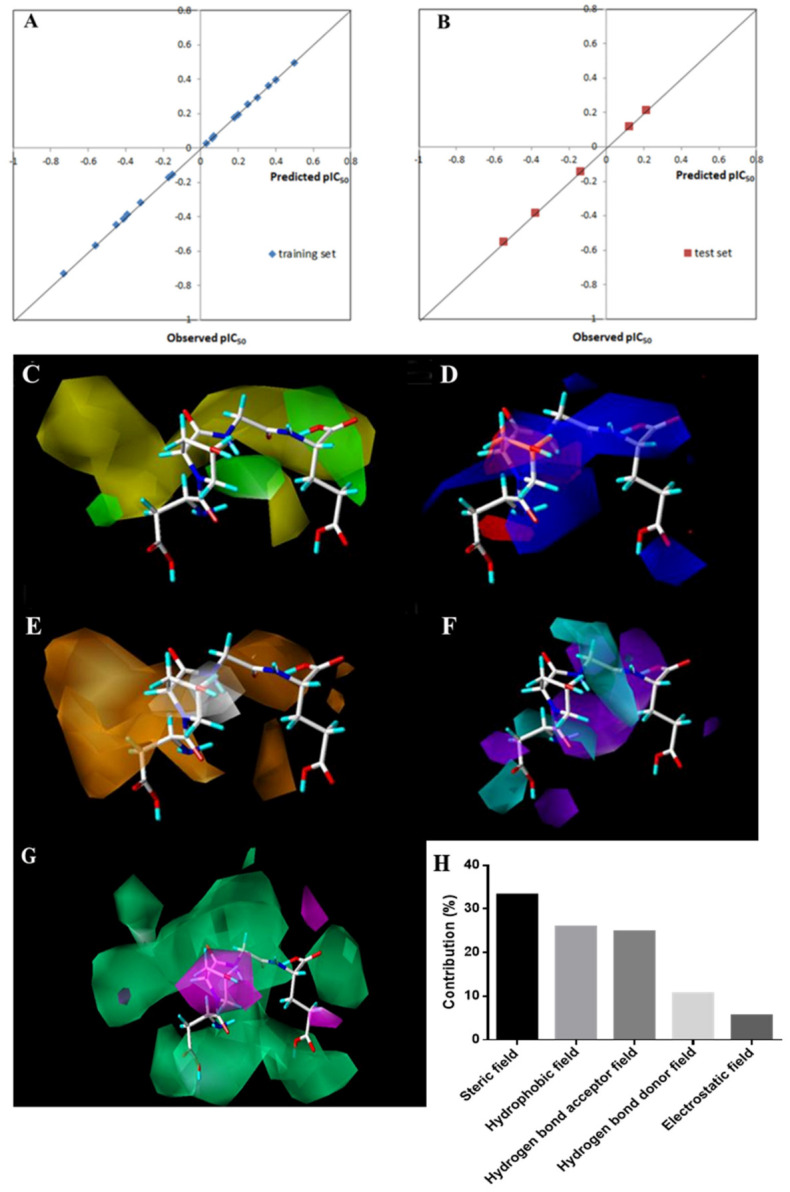
The 3D-QSAR model established by CoMSIA. (**A**,**B**) Plots of predicted values of pIC50 versus actual values for training set (r^2^ = 0.999) and test set (r^2^ = 0.999) of antiplatelet peptides using the CoMSIA model. (**C**–**G**) 3D contour plots of the peptide ROGE of the QSAR model using CoMSIA analysis. (**C**) Steric field; (D) electrostatic field; (**E**) hydrophobic field; (**F**) hydrogen bond donor field; (**G**) hydrogen bond acceptor field. (**H**) Total Contribution of different fields.

**Figure 4 foods-12-00777-f004:**
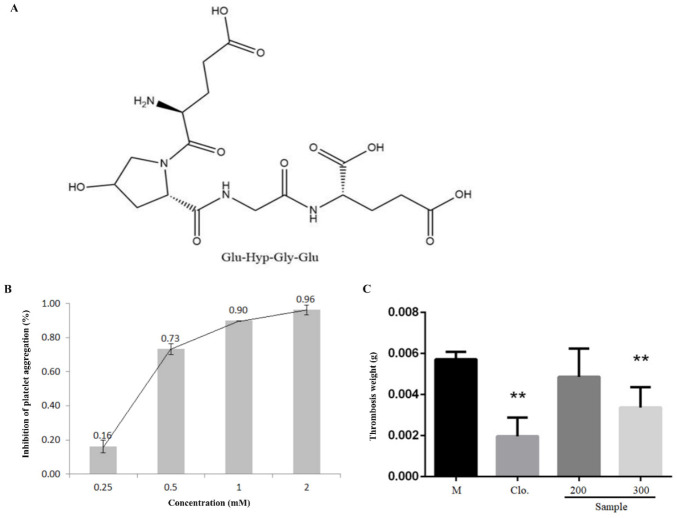
Peptide EOGE exhibited antiplatelet activity in vitro and prevented ferric chloride-induced arterial thrombosis in vivo. (**A**) Chemical structures of peptide EOGE. (**B**) Antiplatelet effect of peptide EOGE induced by ADP. (**C**) Effects of peptide EGOE on ferric chloride-induced arterial thrombosis model in rats, with clopidogrel as positive control. Male Sprague Dawley rats were orally administrated with 0.9% saline, clopidogrel, EGOE at different doses (μmol/kg b.w.), respectively. ** indicates *p* < 0.01 as compared to the M group (*n* = 6).

**Figure 5 foods-12-00777-f005:**
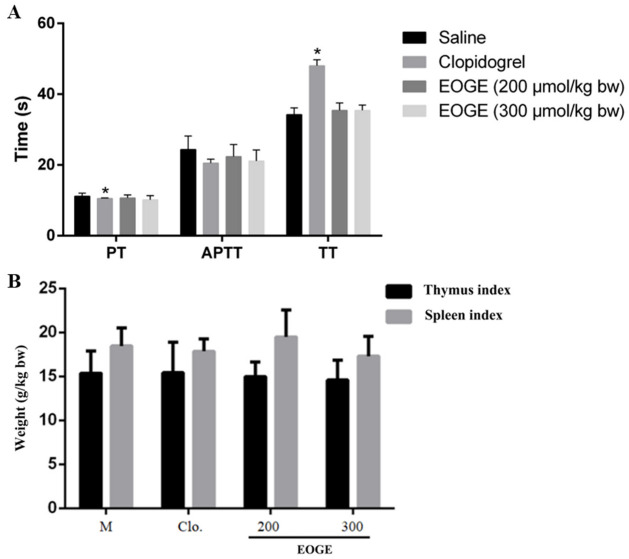
Effects of EOGE on (**A**) PT, APTT, and TT, and (**B**) thymus index and spleen index in rats. Male Sprague Dawley rats were orally administrated with 0.9% saline, clopidogrel, EGOE at different doses (μmol/kg b.w.), respectively. * indicates *p* < 0.05 as compared to the M group (*n* = 6).

**Table 1 foods-12-00777-t001:** Database of antiplatelets for 3D-QSAR models.

No.	Sequence	R1	R2	R3	IC_50_ (mM)	pIC_50_ (mM)	Group
1	PGEOGE	–OH	–Glu–Gly–Pro	–Glu	0.317	0.500	Training
2	EOG	–OH	–Glu	–OH	0.395	0.403	Training
3	OGD	–OH	–H	–Asp	0.435	0.361	Training
4	OGP	–OH	–H	–Pro	0.506	0.296	Training
5	OGSE	–OH	–H	–Ser–Glu	0.558	0.254	Training
6	GOPGPM	–H	–Hyp–Gly	–Pro–Met	5.340	−0.728	Training
7	OGSA	–OH	–H	–Ser–Ala	0.630	0.201	Training
8	DOGE	–OH	–Asp	–Glu	0.661	0.180	Training
9	PG	–H	–H	–OH	3.640	−0.561	Training
10	PGEOG	–OH	–Glu–Gly–Pro	–OH	0.857	0.067	Training
11	GEOG	–OH	–Glu–Gly	–OH	0.878	0.056	Training
12	PGE	–H	–H	–Glu	0.927	0.033	Training
13	PGPK	–H	–H	–Pro–Lys	2.790	−0.446	Training
14	OG	–OH	–H	–OH	1.420	−0.152	Training
15	OGOMG	–OH	–H	–Hyp–Met–Gly	1.487	−0.172	Training
16	GOOGPQ	–OH	–Hyp–Gly	–Pro–Gln	2.076	−0.317	Training
17	VGPOGPA	–OH	–Pro–Gly–Val	–Pro–Ala	2.448	−0.389	Training
18	PGKP	–H	–H	–Lys–Pro	2.571	−0.410	Training
19	PGQP	–H	–H	–Gln–Pro	2.410	−0.382	Test
20	PGPQ	–H	–H	–Pro–Gln	3.565	−0.552	Test
21	PGHH	–H	–H	–His–His	1.397	−0.145	Test
22	OGQ	–OH	–H	–Gln	0.760	0.119	Test
23	OGE	–OH	–H	–Glu	0.620	0.208	Test

**Table 2 foods-12-00777-t002:** Statistical results of QSAR models using Topomer CoMFA and CoMSIA.

Models	q^2^	N	r^2^	r^2^_pred_	SEE	F	Contribution (%)				
							S	E	H	D	A
Topomer CoMFA	0.710	2	0.862	0.930	-	-	-	-	-	-	-
CoMFA	0.282	5	0.950	0.709	0.091	67.769	-	-	-	-	-
CoMSIA	0.461	16	0.999	0.999	0.006	5827.123	33.2	5.5	25.9	10.6	24.8

S, E, H, D, and A indicate steric field, electrostatic field, hydrophobic field, hydrogen bond donor field, and hydrogen bond acceptor field, respectively.

## Data Availability

The data presented in this study are available on request from the corresponding author.

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
