# Peer review of "Structural Insights on Hyp-Gly-Containing Peptides as Antiplatelet Compounds through Topomer CoMFA and CoMSIA Analysis"

_foods, 2023, doi:10.3390/foods12040777_

Round 1

Reviewer 1 Report

This paper investigated Hyp-Gly-containing peptides through Topomer CoMFA and CoMSIA analysis. This paper should be carefully revised, and the comments are as follows:

 1.      Why to choose “Hyp-Gly-containing peptides”? This is suggested to be better described in the introduction section.

2.      Line40: What is the meaning of OG-5? Why is OG-5 referred to?

3.      Line41: “These results reveal that collagen hydrolysates with OG-containing peptides have potential to be developed as an effective diet supplement to prevent the occurrence of thrombotic disease.” If only one or a few peptides have this activity, it seems that this conclusion cannot be well supported. The authors should better explain this conclusion.

4.      Line48-54: The authors briefly introduced the 3D-QSAR model, however, why this model can be used to investigate antiplatelet peptides have not been explained and introduced. The introduction section should be modified.

5.      Line57: What is the meaning of pIC50?

6.      Twenty-three peptides were used for model building. Is the size of the dataset a little small? How to divide the training and test datasets?

7.      Section2.5: The definition of fragments (R1, R2, R3 and common skeleton) should be introduced.

8.      Section2.5: The topological, electrostatic and steric descriptors should be described.

9.      Line149-159: “As shown in Table 1, the optimal number of principal components of the model established by Topomer CoMFA is 2.……” How to get these results from Table1?

10.  Line173-175: What is the value of correlation coefficient?

11.  Line199-227: This section is confused. PG-containing peptide and OG-containing peptide were not well distinguished and introduced in Figure 1, and therefore the related analysis is very confusing to be understood.

12.  What is the value of correlation coefficient in Figure 2A-B?

13.  There is little discussion and few references in the Result and Discussion section of this paper. This should be revised to interpret the results in this study.

Author Response

This paper investigated Hyp-Gly-containing peptides through Topomer CoMFA and CoMSIA analysis. This paper should be carefully revised, and the comments are as follows:

  1. Why to choose “Hyp-Gly-containing peptides”? This is suggested to be better described in the introduction section.

 Response: Collagen has a unique triple-helix structure with a repeated amino acid sequence of (Gly-X-Y)n, in which X and Y are typically Pro and Hyp. Hyp-Gly-containing peptides are abundance in collagen and various Hyp-Gly-containing peptides could be released after hydrolysis. In our previous study, multiple antiplatelet aggregation peptides were identified and Hyp-Gly motif is the common sequence in most peptides. These peptides exhibit potent antiplatelet activity. Therefore, we chose Hyp-Gly-containing peptides to investigate the relationship between structure and activity in this study.

Reference:

[1] Yazaki, M.; Ito, Y.; Yamada, M.; Goulas, S.; Teramoto, S.; Nakaya, M.A.; Ohno, S.; Yamaguchi, K. Oral Ingestion of Collagen Hydrolysate Leads to the Transportation of Highly Concentrated Gly-Pro-Hyp and Its Hydrolyzed Form of Pro-Hyp into the Bloodstream and Skin. J Agric Food Chem 2017, 65, 2315-2322, doi:10.1021/acs.jafc.6b05679.

[2] Tian, Q.; Li, S.M.; Li, B. The Pro-Gly or Hyp-Gly Containing Peptides from Absorbates of Fish Skin Collagen Hydrolysates Inhibit Platelet Aggregation and Target P2Y12 Receptor by Molecular Docking. Foods 2021, 10, 1553, doi:10.3390/foods10071553.

[3] Yang, Y.; Wang, B.; Tian, Q.; Li, B. Purification and Characterization of Novel Collagen Peptides against Platelet Aggregation and Thrombosis from Salmo salar. ACS Omega 2020, 5, 19995-20003, doi:10.1021/acsomega.0c01340.

[4] Song, H.; Tian, Q.; Li, B. Novel Hyp-Gly-containing antiplatelet peptides from collagen hydrolysate after simulated gastrointestinal digestion and intestinal absorption. Food Funct 2020, 11, 5553-5564, doi:10.1039/d0fo00219d.

  1. Line40: What is the meaning of OG-5? Why is OG-5 referred to?

Response: Peptide OG-5 referred to OGEFG. The sequence has been added in Line 48.

  1. Line41: “These results reveal that collagen hydrolysates with OG-containing peptides have potential to be developed as an effective diet supplement to prevent the occurrence of thrombotic disease.” If only one or a few peptides have this activity, it seems that this conclusion cannot be well supported. The authors should better explain this conclusion.

Response: The revision has been made as suggested.

  1. Line48-54: The authors briefly introduced the 3D-QSAR model, however, why this model can be used to investigate antiplatelet peptides have not been explained and introduced. The introduction section should be modified.

Response: The revision has been made as suggested in Line 56-59.

  1. Line57: What is the meaning of pIC50?

Response: pIC50 refers to -logIC50

  1. Twenty-three peptides were used for model building. Is the size of the dataset a little small? How to divide the training and test datasets?

Response: twenty three peptides is the minimum requirement for model building.

The peptides used for modeling are based on existing in natural collagen amino acid sequences, and the number of amino acid is less than 7. Since bigger than octopeptide can not be calculated in our computer. Moreover, the identified sequence with Hyp-Gly motifs in our previous studies were smaller peptides (less than 7). Therefore, we selected and synthesized 10 peptides derived from collagen amino acid sequence, whose inhibitory effect against platelet aggregation were determined. Though the number of training datasets was limited, these peptides showed a good alignment. With more and more OG-containing antiplatelet peptides were identified, the future prediction model will be more accurate and precise.

Peptide No. 19- No. 23 was used in test datasets, which contains low, moderate and high biological activity. Revision has been made in Table 1.

  1. Section2.5: The definition of fragments (R1, R2, R3 and common skeleton) should be introduced.

Response: The definition of R1, R2 and R3 fragments has been added in Line 189-192 as shown in Fig. 1A.

  1. Section2.5: The topological, electrostatic and steric descriptors should be described.

Response: The topological, electrostatic and steric descriptors have described in Line 121-123.

  1. Line149-159: “As shown in Table 1, the optimal number of principal components of the model established by Topomer CoMFA is 2.……” How to get these results from Table1?

Response: These results were from Table 2. The explanation has been added in Line 170. The higher q2, r2 and r2pred and F-values, along with the lower SEE value, indicate a better statistical correlation and reasonable predictability of the QSAR model.

  1. Line173-175: What is the value of correlation coefficient?

Response: The correlation coefficient value has been added in Line 198.

  1. Line199-227: This section is confused. PG-containing peptide and OG-containing peptide were not well distinguished and introduced in Figure 1, and therefore the related analysis is very confusing to be understood.

Response: The structure division of OG-containing peptide has been added in Fig. 1 and the relevant explanation has been added as suggested.

  1. What is the value of correlation coefficient in Figure 2A-B?

Response: The correlation coefficient value has been added in Line 268.

  1. There is little discussion and few references in the Result and Discussion section of this paper. This should be revised to interpret the results in this study.

Response: The revision has been made as suggested in the Discussion section and the references have been added.

Reviewer 2 Report

The authors report some computational and experimental studies to establish 3D-QSAR model to predict the structure features of some antiplatelet peptides.

The topic is interesting, but some improvement can be done. In particular the authors use topomer CoMFA and CoMSIA analysis to establish 3D-QSAR model using peptides already selected in a previous publication.

To verify the predicted features I suggest to perform docking studies with MD refinement od the complexes between selected peptide and their receptor.

Author Response

The authors report some computational and experimental studies to establish 3D-QSAR model to predict the structure features of some antiplatelet peptides.

The topic is interesting, but some improvement can be done. In particular the authors use topomer CoMFA and CoMSIA analysis to establish 3D-QSAR model using peptides already selected in a previous publication.

To verify the predicted features I suggest to perform docking studies with MD refinement od the complexes between selected peptide and their receptor.

Response: In our previous study, the peptides containing Pro-Gly (PG) or Hyp-Gly (OG) sequence were shown to inhibit ADP-induced platelet aggregation in vitro significantly, which is related to the activation of P2Y12 receptor. Molecular docking has been performed and the results implied that the OG-containing peptides might target the P2Y12 receptor and form hydrogen bonds with the key sites Cys97, Ser101, and Lys179.

Reference:

[1] TIAN Q, LI S M, LI B. The Pro-Gly or Hyp-Gly Containing Peptides from Absorbates of Fish Skin Collagen Hydrolysates Inhibit Platelet Aggregation and Target P2Y12 Receptor by Molecular Docking. Foods, 2021, 10(7): 1553.

[2] YANG Y, LIU H, CUI L, et al. A Collagen-Derived Oligopeptide from Salmo salar Collagen Hydrolysates Restrains Atherogenesis in ApoE(-/-) Mice via Targeting P2Y12 Receptor. Mol Nutr Food Res, 2022, 66(13): e2200166.

Reviewer 3 Report

INTRODUCTION:

-          In the first sentence authors state “… in the pathogenesis of numerous cardiovascular diseases (CVD)”. Please provide some examples of these pathologies.

-          Please indicates what Salmo Salar and hypophthalmichthys molitrix are (i.e. fishes).

-          What is OG-5? From what has been extracted?

-          Please provide some information about what CoMSIA is in the introduction.

MATERIALS and METHODS:

-          Paragraph 2.3: Why platelet aggregation was observed at 5 minutes?

-          Paragraph 2.7: What is clopidogrel? Why it has been used in the current study?

-          Do authors have a peptide from Salmo Salar or hypophthalmichthys molitrix that could be used as a control for the experiments? I believe it would be important to be sure that the observed effects are specific for EOGE. If this point has been already published, please make it clear in the current paper.

RESULTS and DISCUSSION:

-          Paragraph 3.4: How was the dose of 300umol/kg of EOGE has been chosen?

-          Do authors have any idea of the possible long term effects (or side effects) of EOGE?

-          Do authors know what could happen after multiple administrations of EOGE?

-          Do authors know if EOGE could work also if given intravenously /intraperitoneally/ subcutaneously?

Author Response

INTRODUCTION:

-          In the first sentence authors state “… in the pathogenesis of numerous cardiovascular diseases (CVD)”. Please provide some examples of these pathologies.

Response: The revision has been made as suggested (Line 31-32).

-          Please indicates what Salmo Salar and hypophthalmichthys molitrix are (i.e. fishes).

Response: The revision has been made as suggested (Line 14).

-          What is OG-5? From what has been extracted?

Response: Peptide OG-5 referred to OGEFG, which was extracted and identified from Salmo Salar skin.

-          Please provide some information about what CoMSIA is in the introduction.

Response: The information of CoMSIA has been added in Introduction section. (Line 62-64)

MATERIALS and METHODS:

-          Paragraph 2.3: Why platelet aggregation was observed at 5 minutes?

Response: The platelet aggregation in vitro experiment is widely used to investigate the antiplatelet aggregation activity of compounds. When platelets are stimulated by agonists, such as ADP or thrombin, the platelets will be activated and the aggregation activity increased with the increase of incubation time. The maximum aggregation activity was recorded at 5 min to calculate the antiplatelet aggregation effect of peptides.

Reference:

[1] YOSHIDA S, SUDO T, NIIMI M, et al. Inhibition of collagen-induced platelet aggregation by anopheline antiplatelet protein, a saliva protein from a malaria vector mosquito. Blood, 2008, 111(4): 2007-14.

[2] SU X L, SU W, HE Z L, et al. Tripeptide SQL Inhibits Platelet Aggregation and Thrombus Formation by Affecting PI3K/Akt Signaling. J Cardiovasc Pharmacol, 2015, 66(3): 254-60.

[3] SEO E J, NGOC T M, LEE S M, et al. Chrysophanol-8-O-glucoside, an anthraquinone derivative in rhubarb, has antiplatelet and anticoagulant activities. J Pharmacol Sci, 2012, 118(2): 245-54.

-          Paragraph 2.7: What is clopidogrel? Why it has been used in the current study?

Response: Clopidogrel is a commercial platelet P2Y12 receptor inhibitor, which could inhibit platelet aggregation to reduce thrombosis. In this study, clopidogrel was used as a positive control group in animal study. (Line 317)

Reference:

[1] SAVI P, HEILMANN E, NURDEN P, et al. Clopidogrel: An Antithrombotic Drug Acting on the ADP-dependent Activation Pathway of Human Platelets. Clinical & Applied Thrombosis/hemostasis, 1996, 2(1): 35-42.

[2] LIU O, JIA L, LIU X, et al. Clopidogrel, a platelet P2Y12 receptor inhibitor, reduces vascular inflammation and angiotensin II induced-abdominal aortic aneurysm progression. PLoS One, 2012, 7(12): e51707.

-          Do authors have a peptide from Salmo Salar or hypophthalmichthys molitrix that could be used as a control for the experiments? I believe it would be important to be sure that the observed effects are specific for EOGE. If this point has been already published, please make it clear in the current paper.

Response: The peptide OGE and OGSA were identified from hypophthalmichthys molitrix and exhibited the highest antiplatelet aggregation activity compared with other identified peptides with IC50 values of 0.62 and 0.63 mM, respectively. EOGE was predicted according to the 3D-QSAR model and was observed to have higher activity with IC50 values of 0.39 mM. However, EOGE is prone to degradation in vivo (data not shown). Consequently, no significant increase of anti-thrombosis effect was observed when compared with that of peptide OGE. The comparison has been added in Line 304-310.

RESULTS and DISCUSSION:

-          Paragraph 3.4: How was the dose of 300umol/kg of EOGE has been chosen?

Response: The administration dose of 300 umol/kg bw was chosen according the effective dose from our previous study, such as peptide ME, OGEFG and DEGP.

Reference:

[1] YANG Y, SONG H, WANG B, et al. A novel di-peptide Met-Glu from collagen hydrolysates inhibits platelet aggregation and thrombus formation via regulation of Gq-mediated signaling. Journal of Food Biochemistry, 2020, 44(9): e13352.

[2] YANG Y, WANG B, TIAN Q, et al. Purification and Characterization of Novel Collagen Peptides against Platelet Aggregation and Thrombosis from Salmo salar. ACS Omega, 2020, 5(32): 19995-20003.

-          Do authors have any idea of the possible long term effects (or side effects) of EOGE?

Response: So far, we have not evaluated the long-term effect of EOGE. But our previous study showed that long-term oral administration of peptide OG-5 for 16-week significantly reduced atherosclerotic plaque formation without side effects in ApoE–/– mice via target P2Y12 receptor, whose mechanism is similar to EOGE. On the other hand, short-term administration of EOGE did not prolong the bleeding time. These results indicate that EGOE may has few side effect for long-term administration.

Reference:

[1] YANG Y, LIU H, CUI L, et al. A Collagen-Derived Oligopeptide from Salmo salar Collagen Hydrolysates Restrains Atherogenesis in ApoE(-/-) Mice via Targeting P2Y12 Receptor. Mol Nutr Food Res, 2022, 66(13): e2200166.

-          Do authors know what could happen after multiple administrations of EOGE?

Response: The present study showed that peptide EOGE was a platelet aggregation inhibitor. Therefore, bleeding risk may be the most common effect after multiple administration of EOGE.

-          Do authors know if EOGE could work also if given intravenously /intraperitoneally/ subcutaneously?

Response: In this study, EOGE was oral administration to animals. As a consequence, plenty of peptide EOGE would be degraded since the gastrointestinal barrier and peptidase hydrolysis in plasma. Our previous study showed that EOGE had a poor bioavailability and is prone to degradation in vivo. If given intravenously /intraperitoneally/ subcutaneously, EOGE may work with more efficient at a lower dose.

Round 2

Reviewer 1 Report

Accept

Reviewer 2 Report

The paper can be accepted for publication.